# A molecular proximity sensor based on an engineered, dual-component guide RNA

**Junhong Choi[1,2]\*, Wei Chen[1,3], Hanna Liao[1,4], Xiaoyi Li[1], Jay Shendure[1,5,6,7,8]\***

[1]Department of Genome Sciences, University of Washington, Seattle, United States; [2]Developmental Biology Program, Memorial Sloan Kettering Cancer Center, New York, United States; [3]Institute for Protein Design, University of Washington, Seattle, United States; [4]Molecular and Cellular Biology Program, University of Washington, Seattle, United States; [5]Howard Hughes Medical Institute, Seattle, United States; [6]Brotman Baty Institute for Precision Medicine, Seattle, United States; [7]Allen Discovery Center for Cell Lineage Tracing, Seattle, United States; [8]Seattle Hub for Synthetic Biology, Seattle, United States

## eLife Assessment

This **important** manuscript describes a creative approach using dual-component gRNAs to create a new class of molecular proximity sensors for genome editing. The authors demonstrate that this tool can be coupled with several different gene editing effectors, showing **convincingly** that the tool functions as intended. This study not only introduces a first-of-its kind approach, but through careful measurements also enables future further development of the technology.

**\*For correspondence:**
choij10@mskcc.org (JC);
shendure@uw.edu (JS)

**Abstract** One of the goals of synthetic biology is to enable the design of arbitrary molecular circuits with programmable inputs and outputs. Such circuits bridge the properties of electronic and natural circuits, processing information in a predictable manner within living cells. Genome editing is a potentially powerful component of synthetic molecular circuits, whether for modulating the expression of a target gene or for stably recording information to genomic DNA. However, programming molecular events such as protein-protein interactions or induced proximity as triggers for genome editing remains challenging. Here, we demonstrate a strategy termed 'P3 editing', which links protein-protein proximity to the formation of a functional CRISPR-Cas9 dual-component guide RNA. By engineering the crRNA:tracrRNA interaction, we demonstrate that various known protein-protein interactions, as well as the chemically induced dimerization of protein domains, can be used to activate prime editing or base editing in human cells. Additionally, we explore how P3 editing can incorporate outputs from ADAR-based RNA sensors, potentially allowing specific RNAs to induce specific genome edits within a larger circuit. Our strategy enhances the controllability of CRISPR-based genome editing, facilitating its use in synthetic molecular circuits deployed in living cells.

## Introduction

Interactions among genes, proteins, and other biomolecules form molecular circuits that process information. Endogenous molecular circuits are composed of different functional modules that can sense, transmit, or integrate various events experienced by cells as inputs. The processed information is used to direct cellular processes, often by altering the expression of specific genes or by inducing chemical modifications to existing molecules (e.g. phosphorylation cascades). One of the goals of the field of synthetic biology is to augment the endogenous molecular circuitry with synthetic components that process-specific input signal(s) to desired output(s) in a predictable fashion (***Wang et al., 2013***). For

**eLife digest** Humans are made up of many building blocks known as cells. The lives of cells are dynamic: they change what tasks they perform over time in response to cues from the rest of the body and the external environment. The mixture of proteins and other molecules present inside a cell, and how they interact with each other, influences how the cell behaves. There are many tools available to take snapshots of these molecules at specific moments, but few technologies that can measure them over periods of time.

A technology known as CRISPR genome editing enables researchers to modify the DNA of cells in a very precise and efficient way. It was adapted from a system that is naturally found in bacteria involving an enzyme called Cas9. Researchers design a molecule known as a guide ribonucleic acid (or guide RNA, for short) that binds to a specific location in the DNA of the cell. Cas9 then binds to the guide RNA and cuts the DNA at this location. When the cell repairs the cut, researchers can manipulate the repair process to make small edits to the DNA, or add or remove larger sections.

Choi et al. set out to develop a new method for recording when molecules within living cells interact using CRISPR-based tools. The records would be in the form of changes to the cells' DNA that could be detected later using existing DNA sequencing technologies. The team split up the CRISPR guide RNA into two parts and attached extra RNA 'adaptors' to enable them to bind to two different proteins of interest. When the two proteins interacted with each other inside human kidney cells, the two halves of the guide RNA were brought together, and this enabled the guide RNA to drive specific editing of the cells' DNA. Choi et al. dubbed this new approach P3 editing.

In the future, it may be possible to combine P3 editing with methods to record other aspects of cell biology into a cell's DNA to reconstruct the history of that cell. One of the next steps following on from this work is to continue developing the P3 editing approach so that it can be more reliably delivered to cells and is more efficient at recording when molecules interact.

example, various synthetic circuits have been built to modulate gene expression or post-translational modifications, in response to a wide range of input signals such as the presence of specific small molecules (*Chen and Elowitz, 2021*).

Genome editing is a potentially attractive output for synthetic molecular circuits. Changes in the genome (or epigenome) can alter the expression of specific genes, unlock the synthesis of specific proteins, or simply serve as a record of past events stably etched into genomic DNA (*Sheth and Wang, 2018*). In nature, targeted alterations of genomic DNA are used to record exposure to specific pathogens (e.g. CRISPR) or to generate diversity in recognition molecules that discriminate self from non-self (e.g. antibodies). In particular, the CRISPR system for immune response in bacteria has been repurposed as a programmable genome editing method (*Cong et al., 2013*; *Jinek et al., 2012*; *Mali et al., 2013*). Since its initial use for genome editing via a programmable nuclease, various genome and epigenome editing methods have been developed that leverage the CRISPR-Cas9 protein as an integral component. Modifications of the Cas9 protein have facilitated more precise genome editing (e.g. base editing, prime editing) or epigenetic control of gene expression (e.g. CRISPRa, CRIPSRi) (*Gilbert et al., 2014*; *Gilbert et al., 2013*).

A critical feature of CRISPR-based genome editing is its straightforward programmability, as the CRISPR guide RNA (gRNA) molecule alone can specifically program the genomic location to be edited. Prime editing (*Anzalone et al., 2019*) extends this advantage via a prime editing guide RNA (pegRNA) that programs both the target locus and editing outcome. However, to fully realize CRISPR's potential in synthetic molecular circuits, we also require machinery that transduces molecular events into genome editing events. For example, the ENGRAM method transduces the output of transcriptional reporters into signal-specific prime editing events (*Chen et al., 2024*). However, it remains unclear how to develop sensors that trigger specific genome editing outcomes for a broader range of event classes (e.g. protein-protein interactions, exposure to small molecules, gene expression, etc.).

RNA engineering is a promising way to construct molecular sensors. RNA aptamers have been developed to sense exposure to small molecules, both outside and inside living cells (*Dykstra et al., 2022*). Similar approaches have been taken to engineer the CRISPR-gRNA to sense small molecules (*Iwasaki et al., 2020*; *Kundert et al., 2019*; *Tang et al., 2017*) or the presence of specific RNA

molecules via RNA-RNA base-pair interactions (*Pelea et al., 2022*; *Wang et al., 2023*). The main advantage of engineering the CRISPR-gRNA instead of protein components is the possibility of multiplexing, i.e., linking each molecular sensor to a different genome editing target site and/or editing outcome. However, both the aptamer-based and RNA-sensing sgRNA strategies are limited by the available functionalities of RNA aptamers in the cellular context. In particular, RNA aptamers or RNA-sensing sgRNAs cannot sense the specific proteins or protein-protein interactions that are at the heart of the molecular circuits underlying most cellular processes.

One of the major functionalities of protein molecules is their ability to recognize and interact with their on-pathway binding partners. The function of a protein might be to simply recognize another peptide with high affinity and specificity as in the case of epitope-antibody interaction, or their interactions can be dynamically controlled by post-transcriptional modifications or the presence of small molecules that act as 'molecular glues' (*Schreiber, 2021*). Using such protein-sensing elements, one can imagine constructing synthetic circuits that sense protein-based signals to conditionally trigger genome editing of a specific genomic location, possibly in the form of a signal-specific editing outcome. The edit could be used to change the function of a specific gene via insertion/deletion/substitution, to modulate its expression via CRISPRa/i, or to record a memory of the signal into the genome via a 'DNA Typewriter Tape' (*Choi et al., 2022*).

Here, we present a strategy named 'P3 editing' (protein-protein proximity), in which specific protein-protein interactions or proximity events promote the formation of the active CRISPR-gRNA complex. P3 editing is based on the dual-component gRNA of the native CRISPR-Cas9 system (*Jinek et al., 2012*), but with the crRNA:tracrRNA dimerization domain replaced with two protein-binding RNA aptamers such as MS2 and BoxB hairpins. Interactions or proximity between two different proteins tagged with MCP (binding MS2 RNA aptamer) and LambdaN (binding BoxB RNA aptamer) domains promote the formation of functional gRNA to induce genome editing. We demonstrate that P3 editing can be used in concert with both base editing and prime editing, converting known protein-protein interactions into genome editing events in human cells. Finally, we explore whether P3 editing can be coupled to ADAR-based RNA sensors to control genome editing based on RNA expression, potentially expanding the range of inputs that can be used to drive signal-specific editing in synthetic molecular circuits.

## Results

### Testing three strategies for leveraging the gRNA as a dimerization module

To achieve a molecular proximity sensor that drives genome editing, a specific physical interaction between two molecules needs to be converted into a successful genome editing event. One possibility is to 'split' a key molecule into two parts, but in such a way that bringing complementary non-functional parts into proximity restores its molecular function. This strategy has been successful in various protein designs (*Michnick et al., 2007*), such as split-GFP (*Ghosh et al., 2000*), split-Luciferase (*Kim et al., 2004*), or split-Protease (*Wehr et al., 2006*) to convert protein-protein interactions or proximities into various output signals based on fluorescence, bioluminescence, or protein modification. In the split architecture, the 'dimerization module' is a key sensor component. Although strategies that split the protein component of the genome editing complex have been described (e.g. split-Cas9 [*Yu et al., 2020*]), we reasoned that having the gRNA serve as the dimerization module rather than the protein, i.e., by splitting it into two parts, and making the restoration of its function dependent on a molecular proximity event, would afford even more control. For example, if multiple split gRNAs were present within the same cell, they could be independently controlled, whereas a split Cas9 would only allow a single control point. In our initial experiments, we focused on splitting the pegRNA used in prime editing.

We considered three broad strategies in designing a 'split-pegRNA' system. First, the pegRNA could be split into a functional sgRNA and its 3'-extension sequence containing the reverse-transcription template and primer binding sequence, the latter also referred to as prime editing trans RNA or petRNA by a recent report (*Liu et al., 2022*; *Figure 1a*, top). Second, an extra sequence motif such as a self-splicing split-ribozyme could be inserted within the pegRNA sequence, such that the coming together of split-ribozyme parts would be required to splice out the extraneous sequence and yield a

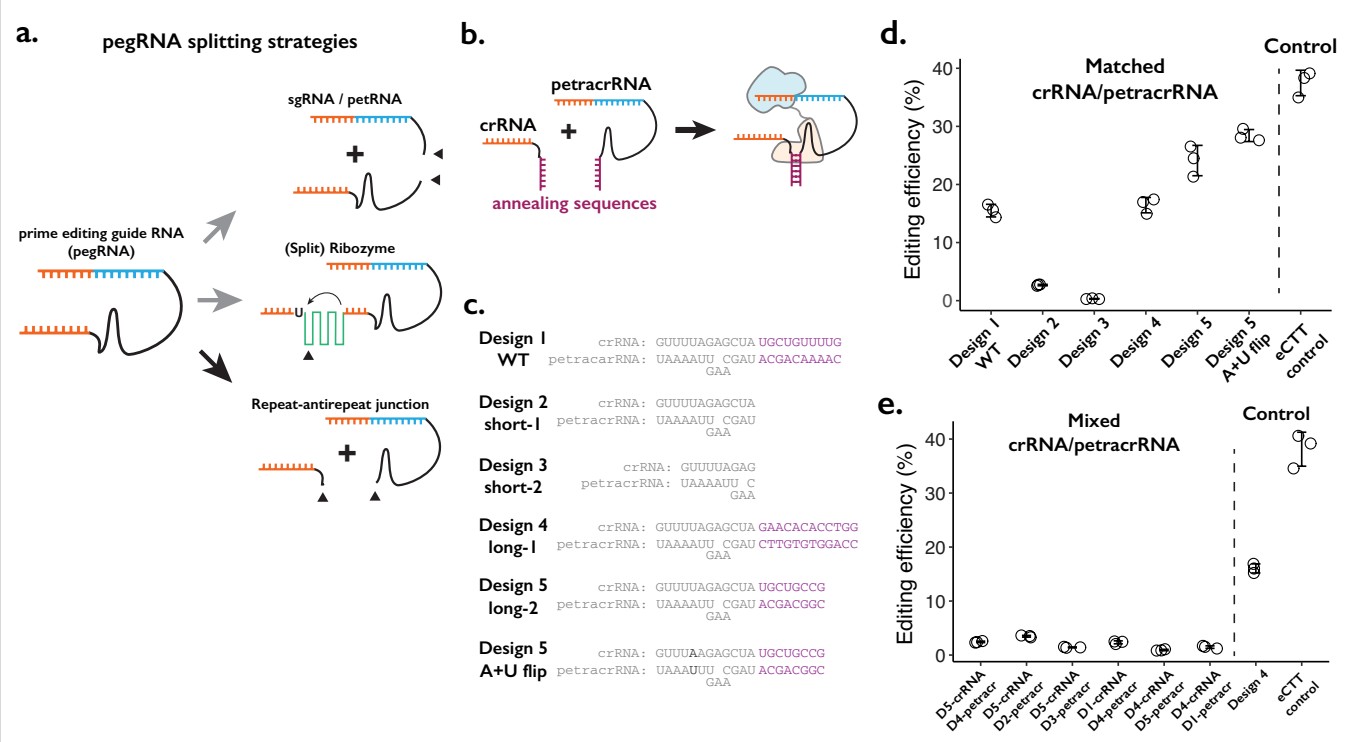

**Figure 1.** Testing the consequences of splitting the prime editing guide RNA (pegRNA) at the repeat:anti-repeat junction. (**a**) We tested three classes of split-pegRNA designs: (Top) pegRNA is split into sgRNA and petRNA. (Center) Self-splicing ribozyme is inserted into pegRNA, thus splitting it into two parts. (Bottom) The Cas9-binding scaffold is split at the crRNA:tracrRNA junction, which is joined through a GAAA RNA tetraloop in the standard sgRNA. (**b**) Dimerization of crRNA and prime editing tracrRNA (petracrRNA) for Cas9 activity is guided by RNA annealing sequences (shown in purple) that are complementary. (**c**) Different designs of complementary sequences for the crRNA-petracrRNA interaction. (**d–e**) Editing efficiencies for prime editing (CTT insertion to HEK3 native genomic locus) using matching (**d**) or mixed (**e**) crRNA/petracrRNA pairs. CTT insertion efficiency using enhanced pegRNA (epegRNA) was used as the positive control ('eCTT control'). The center and error bars are mean and standard deviations, respectively, from n = 3 transfection replicates.

The online version of this article includes the following source data and figure supplement(s) for figure 1:

**Source data 1.** Source data (editing efficiencies) related to *Figure 1* and figure supplements.

**Figure supplement 1.** Testing the sgRNA:petRNA splitting strategy.

**Figure supplement 2.** Testing the self-splicing ribozyme strategy.

functional pegRNA (*Figure 1a*, middle). Third, the pegRNA could be divided at the repeat:anti-repeat junction, which was originally joined with a GAAA RNA tetraloop to form a single-guide RNA (sgRNA) molecule from crRNA and tracrRNA (*Jinek et al., 2012*; *Figure 1a*, bottom).

We first tested splitting the pegRNA into a functional sgRNA and petRNA (*Figure 1a*, top; *Figure 1—figure supplement 1a–b*). We reasoned that the addition of complementary RNA sequences as a 'dimerization module' at the split junction of sgRNA and petRNA would drive the formation of an active pegRNA complex only when the correct complementary sequences are present. To inhibit early degradation of split RNAs, we appended an additional RNA pseudoknot structure at the ends of both molecules, borrowing from the strategy used to form the enhanced pegRNA or 'epegRNA' for higher prime editing efficiency (*Nelson et al., 2022*). We tested a handful of RNA:RNA dimerization sequences that range from 13 to 164 base-pairs, with an expectation that base-pairing between dimerization sequences would drive the formation of active pegRNA molecules. To measure the editing efficiency of the sgRNA:petRNA splitting strategy, we cloned each RNA in separate RNA expression vectors. We programmed crRNA:petracrRNA to target the endogenous *HEK3* locus and insert CTT at the +0 position relative to the nick in combination with the 'PE4max' prime editor (also referred to as PEmax-P2A-hMLH1dn), as this edit has been previously used to benchmark prime editing (*Anzalone et al., 2019*; *Chen et al., 2021*). In general, we observed a low editing efficiency (<2%) by sgRNA-petRNA pairs that are expected to be dimerized by complementary RNA sequences,

even with an additional 3' RNA pseudoknot structure that inhibits RNA degradation from 3'-end (*Nelson et al., 2022*; *Figure 1—figure supplement 1c*). The inefficiency in editing is unlikely to be due to the inserted dimerization domain because a single pegRNA with additional RNA stem-loop structure at the PE-junction exhibited moderate editing efficiency (~10%; pegRNA w/ SL condition in *Figure 1—figure supplement 1c*). The underlying cause might be the inefficient dimerization driven by RNA-RNA duplex formation in the cellular environment without additional protein binding to the RNA duplex, or possible degradation of RNA duplex that lies outside of the Cas9-gRNA complex, unprotected from other factors for binding.

Second, we tested the idea of inserting self-splicing ribozymes within the pegRNA (*Figure 1a*, middle; *Figure 1—figure supplement 2a*). A potential advantage of this strategy is that splicing out of the ribozyme sequence would result in an identical molecule to the standard pegRNA. We tested six sites within pegRNA to insert the whole self-splicing ribozyme sequence from *Tetrahymena thermophila* (413 bp in length) (*Herschlag and Cech, 1990*; *Figure 1—figure supplement 2b*). If this proved successful, we envisioned that the ribozyme could be split into two parts and function as a 'split-ribozyme' (*Gambill et al., 2022*). However, editing using pegRNAs containing the self-splicing ribozyme sequence was inefficient (<2%) in all six constructs. To the limited extent that editing was observed, we found that it was dependent on ribozyme function because including a loss-of-function substitution in the ribozyme sequence reduced editing efficiency by greater than 10-fold (*Figure 1—figure supplement 2c*). Our results suggest that while prime editing depends on the self-splicing function of the ribozyme to produce an active pegRNA, the overall ribozyme efficiency is too low, possibly due to the presence of a prime editing enzyme that might bind and interfere with self-splicing.

Finally, we tried splitting the pegRNA into a crRNA component and a tracrRNA component with the 3'-extension necessary for prime editing (we term the latter as the prime editing tracrRNA or 'petracrRNA') (*Figure 1a*, bottom; *Figure 1b*). Previous reports have shown that the distal double-stranded RNA region of the crRNA:tracrRNA junction (also known as the repeat:anti-repeat junction) is not necessary for Cas9 function; this region is replaced by the GAAA tetraloop sequence in the standard sgRNA constructs (*Jinek et al., 2012*). Therefore, we replaced this sequence with other complementary RNA sequences and appended an additional RNA pseudoknot structure at the end of both molecules to inhibit early degradation of split RNAs, similar to our testing of sgRNA/petRNA above.

We cloned six pairs of crRNA-petracrRNA pairs, varying sequences that are likely to drive dimerization between crRNA and petracrRNA, but less likely to be essential for interaction between the pegRNA and prime editor (*Figure 1c*). Using the 10 bp dimerization sequence found in the standard sgRNA, we observe editing efficiency at *HEK3* of 15% (*Figure 1d*), which was 37% of the editing efficiency of the standard epegRNA programming of the same edit ('eCTT control') with the same PE4max prime editor. When different dimerization sequences were placed on crRNA and petracrRNA, the editing efficiency varied between 15% and 25%. The editing efficiency was further enhanced to 28% by flipping A-U pairs at the bottom of repeat:anti-repeat duplex, which is 76% of the editing efficiency of the eCTT control. In contrast, removing the dimerization sequence reduced the editing efficiency to 2.7%, which further reduced to 0.3% when the upper part of the Cas9-binding duplex was removed (*Figure 1c and d*; short-1 and short-2 designs). To verify that the observed editing was dependent on RNA-RNA hybridization, if we paired crRNA and petracrRNA with unmatching dimerization sequences, we consistently observed low editing efficiency (2–3%) (*Figure 1e*).

In summary, we tested three strategies to split the pegRNA into two molecules (*Figure 1a*). Among these, splitting the pegRNA at the repeat:anti-repeat portion is the most promising strategy, and can be potentially extended to the general gRNA used with CRISPR-Cas9 (*Figure 1a*, bottom; *Figure 1b–e*). The resulting rates of prime editing are strongly dependent on the complementarity between the repeat:anti-repeat region of crRNA and petracrRNA. Furthermore, our design mimics the functional molecules within the native CRISPR-Cas9 system, initially discovered as the dual-RNA-guided system (*Jinek et al., 2012*). Therefore, we pursued this strategy in further developing a molecular proximity sensor that drives genome editing.

## Controlling genome editing with protein-protein proximity

To convert protein-protein interaction events into genome editing events, we reasoned that the RNA dimerization domain within crRNA and tracrRNA could be replaced with an RNA-based protein dimerization domain, such that specific protein-protein interactions or prolonged proximity would

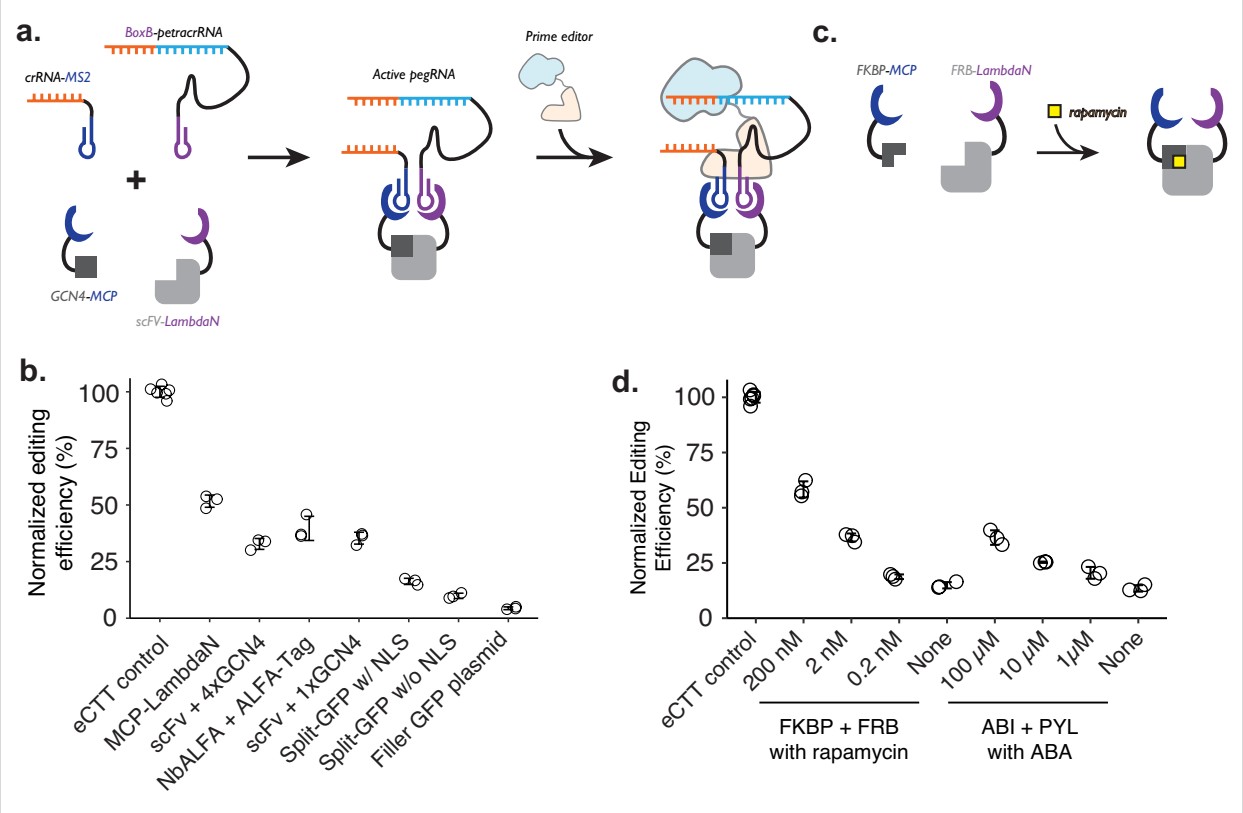

**Figure 2.** Controlling genome editing with constitutive or chemically induced protein-protein interactions. (**a**) Schematic of P3 editing, which couples protein-protein interactions to genome editing. The binding of two proteins (e.g. MCP-tagged GCN4 epitope and LambdaN-tagged scFv) brings two RNA species (e.g. crRNA-MS2 and BoxB-petracrRNA) into close proximity to form an active prime editing guide RNA (pegRNA) complex, which binds to the prime editor for genome editing. (**b**) Normalized editing efficiencies measured with different protein pair interactions. Editing efficiencies were normalized to a consistent eCTT control included in each experiment (left-most). Transfection of filler GFP-expressing plasmid instead of proteins tagged with MCP/LambdaN was used to quantify the background levels of editing as a negative control (right-most). (**c**) Using chemically inducible dimerization to control genome editing. For example, FKBP and FRB domains dimerize in the presence of the small molecule rapamycin. (**d**) Normalized editing efficiencies measured for chemically induced genome editing. The center and error bars are mean and standard deviations, respectively, from n = 3 transfection replicates for both panels c and d.

The online version of this article includes the following source data and figure supplement(s) for figure 2:

**Source data 1.** Source data (editing efficiencies) related to *Figure 2* and figure supplements.

**Figure supplement 1.** Characterizing linkers for crRNA-MS2 and BoxB-petracrRNA.

aid in the formation of active gRNA that would then bind to Cas9 (*Figure 2a*). To date, many specific protein-RNA interaction parts have been identified, including the MS2 RNA aptamer that binds to the MCP domain (*Johansson et al., 1997*). Among different RNA-protein pairs, we chose two pairs, MS2-MCP and BoxB-LambdaN (*Katrekar et al., 2022*; *Legault et al., 1998*; *Salstrom and Szybalski, 1978*), which can be used as adaptors for using protein-protein interactions to drive RNA-RNA interactions.

We designed a crRNA and petracrRNA pair, for which previously identified dimerization sequences of the upper portion of the repeat:anti-repeat duplex were replaced with MS2 and BoxB RNA aptamers. To aid the formation of the proper bulge within the RNA duplex (A and AAG in crRNA and petra-crRNA, respectively), we replaced the original sequence of GCUA in crRNA with GCGC, and UAGC in petracrRNA with GCGC. Finally, we added evoPreQ1 RNA pseudoknot sequence on both RNA molecules to reduce RNA degradation from 3'-ends, resulting in crRNA-MS2 and BoxB-petracrRNA.

To test whether crRNA and petracrRNA modified with protein-binding aptamer sequences can be used to drive genome editing, we constructed a single protein where the MCP and LambdaN domains were fused with a flexible linker (MCP-LambdaN), with the intent of bringing crRNA-MS2 and BoxB-petracrRNA into close proximity to form active pegRNA. When we transfected HEK293T cells with plasmids expressing MCP-LambdaN, PE4max (*Chen et al., 2021*), crRNA-MS2, and BoxB-petracrRNA

that target *HEK3* locus and program for CTT insertion at +0 position, we observed 15% editing efficiency, which was 52% of the editing rate achieved by transfecting a single, standard epegRNA-expressing plasmid that programs the same edit along with the PE4max-expressing plasmid ('eCTT control') (*Figure 2b*). When the MCP-LambdaN-expression plasmid was substituted with a GFP-expressing filler plasmid ('Filler GFP plasmid' condition in *Figure 2b*), editing efficiency dropped to 6% of the eCTT control, indicating that crRNA-MS2 and BoxB-petracrRNA can promote genome editing to a limited extent, presumably by forming an active pegRNA complex even without the additional dimerization domain.

We next sought to extend this approach to link arbitrary protein-protein interactions to genome editing. For this, we designed protein components that can induce the formation of pegRNA complexes containing crRNA-MS2 and BoxB-petracrRNA. As above, because the editing efficiency can vary depending on the transfection efficiency (30–60% depending on the culturing condition and confluency of the cell culture), we continued to use a 'normalized editing efficiency' scale where the observed editing efficiency is scaled to the eCTT control to facilitate comparison across experiments.

We selected three known pairs of interacting protein domains and appended MCP and LambdaN domains with several nuclear localization sequences (*Chen et al., 2021*): (1) GCN4 epitope sequence (20 AA) and single-chain variable fragment (*Colby et al., 2004*; *Lecerf et al., 2001*; *Tanenbaum et al., 2014*; *Wörn and Plückthun, 2001*) (scFv; 250 AA) that recognizes GCN4 epitope, (2) ALFA-tag sequence (15 AA) and nanobody specific for ALFA-tag (NbALFA; 135 AA) (*Götzke et al., 2019*), and (3) split-GFP (*Cabantous et al., 2005*; *Ghosh et al., 2000*), where 11 beta-sheets of GFP is split into GFP1-10 (220 AA) and GFP11 (17 AA). For the GCN4 epitope design, we constructed two designs where either a single repeat of the epitope (1xGCN4) was used, or four repeats of the epitope (4xGCN4) were strung together, as often done to increase the efficiency of scFv binding to the tagged protein molecule. For the split-GFP design, we also constructed a version without any nuclear localization sequences, to test whether protein localization also affects editing efficiency. Designed proteins were cloned into protein expression plasmids and transfected into cells along with crRNA/petra-crRNA/prime editor expressing plasmids.

First, we observe that both epitope-antibody/nanobody interactions were able to induce strong genome editing efficiencies, close to the levels observed with the MCP-LambdaN fusion protein (*Figure 2b*). Interestingly, the additional GCN4 epitopes in the 4xGCN4 design did not seem to increase editing efficiency, possibly meaning either that GCN4:scFv binding is saturated and additional GCN4 epitopes do not make a difference in bringing crRNA and petracrRNA into proximity, or that only the scFv binding to one of GCN4 epitope contributes to the active pegRNA formation due to spatial constraints. Second, we observe that split-GFP designs generally have lower editing efficiency, possibly due to weaker interaction between GFP1-10 and GFP11 compared to epitope-antibody/nanobody interactions. Finally, all three protein pairs as well as MCP-LambdaN fusion constructs include nuclear localization sequences to enhance genome editing in the nucleus. We also tested the split-GFP construct without a nuclear localization sequence, which resulted in an even lower editing efficiency, possibly indicating that the nuclear localization propensity of proteins tagged with the MCP or LambdaN could affect the effectiveness of P3 editing.

Lastly, we tested whether this strategy could be adapted to facilitate small molecule-based control of genome editing. In the past 30 years, several chemicals have been identified as critical signaling molecules for promoting protein-protein interactions (*Schreiber, 2021*). We reasoned that the addition of such chemicals to cell culture could be used to control genome editing of specific targets and editing outcomes. To demonstrate the chemical control of P3 editing, we chose rapamycin-induced dimerization of FKBP and FRB protein domains of the mTOR pathway (*Banaszynski et al., 2005*; *Bierer et al., 1990*; *Brown et al., 1994*) from human proteome, and abscisic acid (ABA)-induced dimerization of pyrabactin resistance domain (PYL) and ABA-insensitive (ABI) domain found in plant (*Liang et al., 2011*; *Figure 2c*). We observed a strong increase in the editing efficiency of the FKBP-FRB and PYL-ABI pairs upon the addition of small molecules that induce dimerization (*Figure 2d*). For example, we observed 3.9-fold higher editing upon the addition of 200 nM rapamycin to the FKBP-FRB pair compared to no rapamycin condition (58% vs 15% normalized editing efficiency), and 2.7-fold higher editing upon the addition of 100 μM ABA to the PYL-ABI pair compared to no ABA condition (36% vs 13% normalized editing efficiency) (*Figure 2d*).

## Optimizing the efficiency vs specificity tradeoff of P3 editing

Overall, these experiments demonstrate that P3 editing can be used to control genome editing via specific interactions between a pair of tagged protein domains. Next, we sought to improve the efficiency and specificity of P3 editing by engineering the crRNA-MS2/BoxB-petracrRNA design. We designed 12 pairs of crRNA-MS2/BoxB-petracrRNA guides, varying the base composition and length of the upper 4 bp region within the Cas9-binding region of repeat:anti-repeat duplex (*Figure 2— figure supplement 1*). Each RNA design was cloned into an RNA expression vector with flanking U6 promoter and TTTTTTT terminator sequences. We transfected HEK293T cells with the mix of four plasmids expressing each component: crRNA-MS2, BoxB-petracrRNA, PE4max, and either split-GFP (GFP1-10 tagged with MCP and GFP11 tagged with LambdaN; to measure the editing efficiency induced by protein-protein interaction) or standard GFP (to measure the background level of editing without protein-protein interaction).

Across 12 designs, we observe a tradeoff between efficiency (ranging between 2% and 53% of normalized eCTT control) and specificity (background editing level without the addition of tagged protein pair ranging between 0.3% and 35%). The efficiency increases with a higher number of G-C

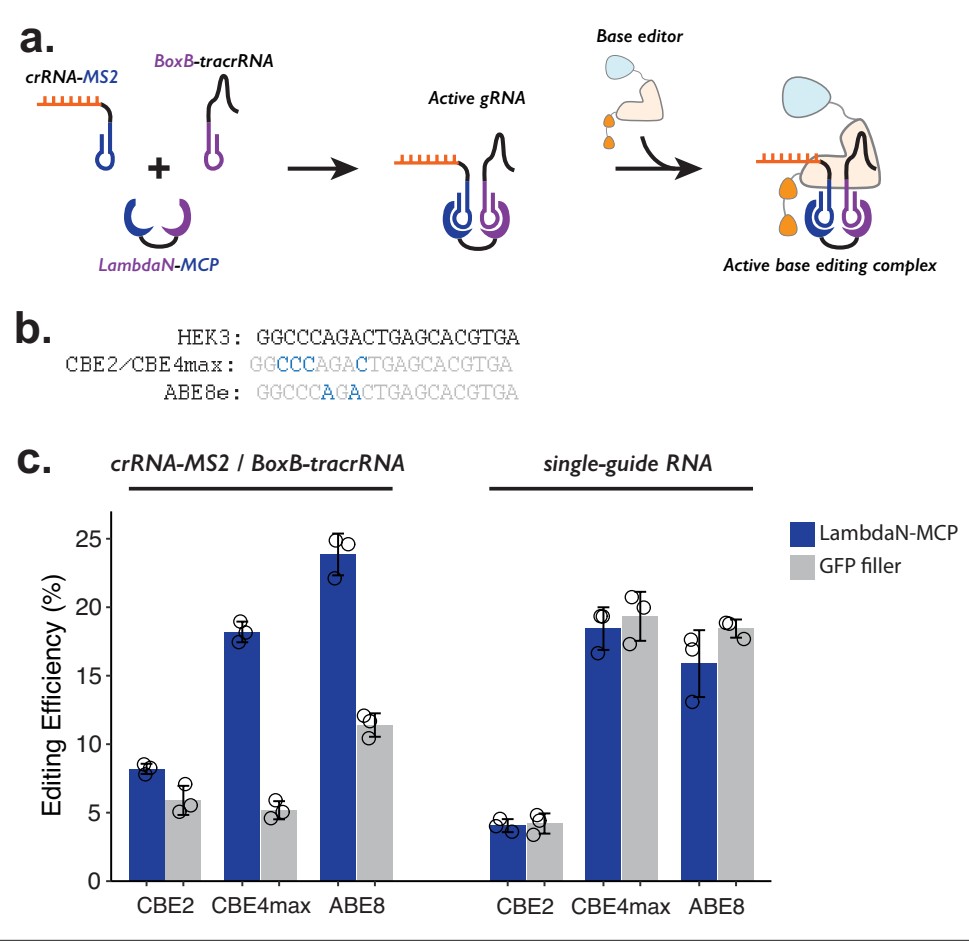

**Figure 3.** Testing P3 editing strategy in the context of base editing. (**a**) Schematics of P3 editing strategy adapted to base editing. The LambdaN-MCP fusion protein will bring crRNA-MS2 and BoxB-tracrRNA in close proximity to aid gRNA complex formation, which then binds to the base editor for genome editing. (**b**) Cytosine and adenine residues targeted by base editing are highlighted within the target HEK3 sequence. (**c**) Editing efficiency measured with P3 editing (left) and standard sgRNA (right). For each base editor and guide-RNA combination, editing efficiencies were measured with either LambdaN-MCP expressing plasmid (blue) or a filler plasmid expressing GFP (gray). The center and error bars are mean and standard deviations, respectively, from n = 3 transfection replicates.

The online version of this article includes the following source data for figure 3:

**Source data 1.** Source data (editing efficiencies) related to *Figure 3* and figure supplements.

base-pairs and annealing length within the varied region, supporting our hypothesis that crRNA-MS2 and BoxB-petracrRNA can form an active pegRNA complex without the addition of protein-mediated proximity, although at a lower rate. However, as the efficiency of P3 editing decreases, the specificity of P3 editing to protein-protein interaction can increase. For example, in our design that uses GAUA:UAUC pairing, we observe 3.8 ± 1.0% normalized editing efficiency with the addition of tagged split-GFP protein, and only 0.27 ± 0.07% normalized editing efficiency without it, a 14-fold increase driven by specific protein-protein interaction. Our optimization effort suggests that the efficiency and specificity of P3 editing can be tuned using different designs of crRNA-MS2/BoxB-petracrRNA, depending on the experimental goal.

## Coupling various modes of genome editing to protein-protein proximity

While our demonstrations so far coupled the P3 strategy to prime editing, we reasoned that the same strategy could be adapted to other precision editing methods such as base editing (*Figure 3a*). To confirm this, we tested three different base editors (CBE2 [*Komor et al., 2016*], CBE4max [*Koblan et al., 2018*], and ABE8 [*Richter et al., 2020*]), which can target the HEK3 locus and make either C-to-T or A-to-G edits (*Figure 3b*). To facilitate compatibility with base editing, we designed BoxB-tracrRNA that lacks the 3'-extension specific to prime editing and cloned it into an RNA expression vector with a U6 promoter. We transfected HEK293T cells with a mix of plasmids expressing: crRNA-MS2, BoxB-tracrRNA, one of the three base editors, and either the LambdaN-MCP fusion protein that brings crRNA-MS2 and BoxB-tracrRNA into close proximity or GFP to serve as a negative control and measure the base-level editing efficiency without a protein component that aids the formation of gRNA complex.

For all three base editors tested, we observe that the addition of LambdaN-MCP increases the editing efficiency (*Figure 3c*, left), although the change in the editing was greater for CBE4max (3.5-fold increase) and ABE8 (2.1-fold increase) than CBE2 (1.4-fold increase). The observed differences are possibly related to different endonuclease activity, as CBE2 uses deactivated Cas9 instead of Cas9 nickase as in the case of the prime editor and other base editors. Of note, the observed genome editing efficiencies of the P3 strategy for all three base editors (*Figure 3c*, left) are comparable to our base editing efficiencies measured with standard sgRNA (*Figure 3c*, right), despite the fact that the P3 strategy uses a three-component system (using crRNA-MS2, BoxB-petracrRNA, and LambdaN-MCP) instead of one sgRNA.

## Combining ADAR-based editing with P3 editing to control genome editing with RNA expression

In both prime editing and base editing demonstrations of the P3 strategy, we used the presence of LambdaN-MCP fusion protein to control the genome editing efficiency, forming a synthetic circuit with the input of protein expression and the output of genome editing. Recently, a similar synthetic circuit was developed using ADAR-based RNA editing, where the input of specific target RNA expression can be used to control an output of cargo protein expression. In this system (termed CellREADR [*Qian et al., 2022*], RADAR [*Kaseniit et al., 2023*], or RADARS [*Jiang et al., 2023*]), an ADAR-guide-RNA is specifically designed to bind the target RNA molecule and promote an RNA base editing event that converts an internal stop codon into a sense codon within the ADAR-guide-RNA, thus initiating cargo protein expression.

To combine the ADAR-based RNA sensor and P3 editing concepts, we programmed the cargo protein downstream of the ADAR-guide-RNA to be a fusion protein (e.g. LambdaN-MCP) that is capable of bringing crRNA and petracrRNA in close proximity (*Figure 4a*). We based our design on the RADARS strategy, which includes an MS2 RNA structure within the ADAR-guide-RNA to aid the recruitment of MCP-tagged ADAR and suppress translation read-through of an internal stop codon. To avoid complications of using MS2 RNA elements in both ADAR-guide-RNA and crRNA, we switched out the MS2/MCP pairing with the PP7 RNA element that binds to the PCP protein (*Chao et al., 2008*; *Olsthoorn et al., 1995*; *Wu et al., 2014*) domain, and confirmed that this could support P3 editing (*Figure 4b*). We used an ADAR-guide-RNA design that detects IL6 mRNA expression from the RADARS system (*Jiang et al., 2023*) because the efficiency and specificity of RNA detection have

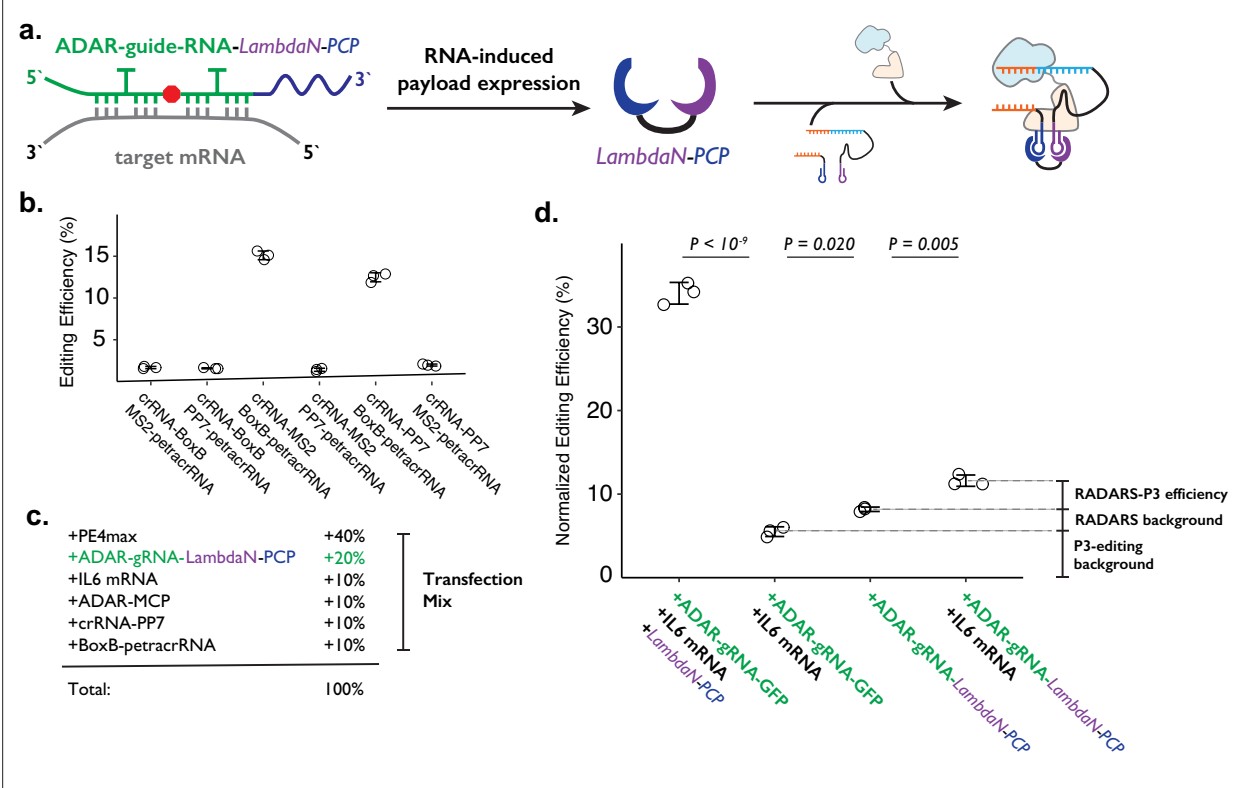

**Figure 4.** Combining ADAR-based RNA sensors with P3 editing. (**a**) Schematics of combining ADAR-based RNA sensors with P3 editing. ADAR-guide-RNA senses target RNA by forming extended RNA duplex based on reverse-complementary sequence, which is a substrate for RNA editing by ADAR. RNA editing converts the in-frame stop codon into a sense codon, which then starts the expression of cargo protein. In combination with P3 editing, the cargo protein will be LambdaN-PCP, which will then help prime editing complex formation by bringing crRNA-PP7 and BoxB-petracrRNA together. (**b**) The six pairs of crRNA/petracrRNA with three different RNA aptamers (BoxB, MS2, and PP7) were tested for P3 editing. Pairing of crRNA-PP7 and BoxB-petracrRNA is similarly compatible with P3 editing as pairing of crRNA-MS2 and BoxB-petracrRNA, and is preferable in this context as it avoids using the MS2 RNA element which is used in RADARS strategy for the ADAR-based RNA sensing. The center and error bars are mean and standard deviations, respectively, from n = 3 transfection replicates. (**c**) Composition of transfection condition with six expression plasmid. The percentage is based on the mass amount of plasmid added to the transfection mix. (**d**) The normalized editing efficiency was measured to characterize the efficiency and specificity of ADAR-based RNA sensing and P3 editing strategy. p-Values were obtained using a two-tailed Student's t-test with Bonferroni correction. The center and error bars are mean and standard deviations, respectively, from n=3 transfection replicates.

The online version of this article includes the following source data for figure 4:

**Source data 1.** Source data (editing efficiencies) related to *Figure 4* and figure supplements.

been previously well characterized in the context of the cargo protein expression of luciferase and GFP.

In combining RADARS with P3 editing, we transfected HEK293T cells with a mix of six plasmids that express: MCP-ddADAR, ADAR-guide-RNA-LambdaN-PCP (also referred to as ogRNA in RADARS), IL6 RNA for target RNA (specifically chosen in RADARS system because it is not natively expressed in HEK293T cells) crRNA-PP7, BoxB-petracrRNA, and PE4max (*Figure 4c*). We observed the normalized editing efficiency of 11.6 ± 0.7%, which was significantly higher than a condition where the IL-6 RNA-expressing plasmid has been replaced with a filler GFP-expression plasmid (8.2 ± 0.3%; p=0.005, where p-values were obtained using the two-tailed Student's t-test with Bonferroni correction) (*Figure 4d*).

While this result suggests that we can control genome editing with specific RNA expression events by combining RADARS and P3 editing, we investigated what contributes to the high background level as well as the low editing efficiency. First, when we replace the plasmid expressing ADAR-guide-RNA-LambdaN-PCP with the ADAR-guide-RNA-GFP (with the cargo protein of GFP instead of LambdaN-PCP), we observe the normalized editing efficiency of 5.5 ± 0.6% (*Figure 4d*). This suggests that the crRNA-PP7 and BoxB-petracrRNA can induce 5.5% normalized editing efficiency without

LambdaN-PCP in the system, characterizing the non-specific editing level of P3 editing, and suggesting that the remaining 2.7% of normalized editing efficiency background is due to non-specific editing of the RADARS system, possibly due to leaky expression of LambdaN-PCP even in the absence of IL6 target RNA. Finally, when we replace 50% of ADAR-guide-RNA-GFP expression plasmid with LambdaN-PCP expression plasmid, we observe the normalized editing efficiency to increase up to 34 ± 1%, suggesting that the inclusion of the RNA-controlled protein expression module might be lowering the overall editing efficiency. Overall, these results suggest that while the P3 editing strategy can be combined with other synthetic circuits to control genome editing with various inputs, the efficiency and specificity of control may degrade over combinations of multiple synthetic modules, and that further optimization is necessary.

## Discussion

Molecular sensors expand our ability to capture and observe biological events as they unfold within living cells (*Chen and Elowitz, 2021*; *Gordley et al., 2016*; *Slusarczyk et al., 2012*). Key parameters to consider in using molecular sensors include their efficiency and specificity in converting events-of-interest into measurable output. Although various sensors of RNA or protein production have been developed, the readout is almost always in the form of luminescence or fluorescence (e.g. luciferase or GFP in the case of RADARS [*Jiang et al., 2023*]), because microscopy- or flow-sorting-based detection methods are common modes of detection with high sensitivity, possibly overcoming low conversion efficiency by sensors as long as they have high specificity.

For several reasons, genome editing is an attractive alternative for recording the output of molecular sensors (*Chen et al., 2024*; *Farzadfard et al., 2019*; *Sheth and Wang, 2018*; *Tang and Liu, 2018*). First, massively parallel sequencing is a low-cost means of acquiring data, and any low conversion efficiency could potentially be overcome simply through deeper sequencing. Second, single-cell resolution can be achieved by coupling recovery of genome editing events with methods such as single-cell RNA-seq. This affords the potential to recover measurements from much larger numbers of cells than would be possible with microscopy or flow, and moreover to co-assay other classes of information about each cell. Third, information encoded within the genome is maintained and replicated throughout the life cycle of the cell, allowing storage of past information with minimal information loss. On a related point, strategies such as DNA Typewriter (*Choi et al., 2022*) can facilitate the time-resolved recording of the output of multiple sensors to a common location(s) in the genome. Finally, molecular sensors can be used to control cellular function as components of synthetic biological circuits. Outputting molecular sensors to genome editing events may facilitate the design of such control structures, particularly as the range of cellular activities that can be programmed by genome editing is rapidly expanding.

In P3 editing, molecular interactions are sensed and converted directly into a genome editing output. Here, we demonstrated that specific RNA-RNA (e.g. annealing of two sequences within crRNA and petracerRNA), RNA-protein (e.g. MS2:MCP and BoxB:LambdaN interactions), and protein-protein (epitope:antibody or chemically induced dimerization) interactions can be used to control genome editing. Because the input of P3 editing is at the protein level, one could also chain different sensors with protein outputs upstream (*Chen and Elowitz, 2021*), although the conversion efficiency and specificity may be compounded and result in higher noise in capturing signals through more relays. The protein-level input also circumvents a key shortcoming of existing signal-specific molecular recording systems, such as DOMINO (*Farzadfard et al., 2019*), CAMERA (*Tang and Liu, 2018*), and ENGRAM (*Chen et al., 2024*), which are limited to recording signal-driven transcription of the gRNA and/or the genome editor. Finally, the sensor elements in the P3 editing strategy are focused on engineering each CRISPR-gRNA while leaving the gene editor (i.e. prime editor and base editor) unaltered, potentially facilitating multiplex genome editing within the same cell. For example, one could imagine developing a P3 editing strategy to construct a sensor specific to a cell state and combine it with other molecular signal or lineage recording methods (*Chen et al., 2024*; *Choi et al., 2022*) to reconstruct the past history of each cell.

In our view, there are four outstanding challenges for P3 editing to be broadly useful: evaluating additional cellular contexts, the method's efficiency and specificity, understanding the limit of detectable protein-protein interactions, and the development of sensors compatible with multiplex P3 editing within the same cell. First, we have thus far only conducted P3 editing in HEK293T cells, and

obviously needs to be tested in additional cell types. Second, both the efficiency and specificity of the P3 editing need to be improved before it can be used as a selective editing tool in model systems. We have explored how modifying the crRNA and petracrRNA pair sequences can tune the efficiency-vs-specificity tradeoff, but alternative avenues to improvement (e.g. better docking of RNA aptamers such as MS2, BoxB, or PP7 by testing more linker sequences that place crRNA and petracrRNA for duplex formation) may be more fruitful in terms of achieving high efficiency and specificity at once (e.g. >50% editing in the setting of a specific protein-protein interaction, and <1% editing without it). Second, it is not clear whether weak and transient interactions among proteins can be used to trigger P3 editing. Assuming the genome editing complex formation is reversible, improving P3 editing efficiency may be able to capture different strengths of protein-protein interactions, although some interactions may be too transient to promote functional gRNA formation. Finally, the current P3 editing design uses a pair of RNA aptamers and their corresponding protein binders, limiting the multiplex detection of protein-protein pairs. More orthogonal protein-RNA pairs need to be identified e.g. using a massively parallel platform (*Buenrostro et al., 2014*) and/or computational prediction (*Baek et al., 2024*) to allow for large numbers of P3 sensors for different protein-protein interactions to be deployed within the same cell. Overcoming these four challenges is necessary for P3 editing to be broadly useful for gating genome editing on physiological levels of specific protein-protein interactions in a multiplex fashion.

# Materials and methods

**Key resources table**

| Reagent type (species) or resource | Designation | Source or reference | Identifiers | Additional information |
|---|---|---|---|---|
| Strain, strain background (*E. coli*) | NEB C3040H Competent cells | NEB | C3040H | |
| Cell line (Homo-sapiens) | HEK293T | ATCC | CRL-3216 | |
| Transfected construct (synthetic) | PE4max | Addgene | Addgene_174828 | |
| Transfected construct (synthetic) | pU6-pegRNA-GG-acceptor | Addgene | Addgene_132777 | |
| Transfected construct (synthetic) | pU6-crRNA-MS2 | Addgene | Addgene_207624 | |
| Transfected construct (synthetic) | pU6-BoxB-petracrRNA | Addgene | Addgene_207625 | |
| Transfected construct (synthetic) | pCMV-LambdaN-MCP | Addgene | Addgene_207626 | |
| Transfected construct (synthetic) | pCMV-LambdaN-NbALFA | Addgene | Addgene_207627 | |
| Transfected construct (synthetic) | pCMV-ALFA-MCP | Addgene | Addgene_207628 | |
| Commercial assay or kit (cloning) | T4 DNA ligase | NEB | M0202S | |
| Commercial assay or kit (cloning) | BsaI-HF-v2 | NEB | R3733S | |
| Commercial assay or kit (cloning) | NEBuilder HiFi DNA assembly master mix | NEB | E2621S | |
| Commercial assay or kit (plasmid purification) | Qiagen miniprep | Qiagen | 27106 | |
| Commercial assay or kit (transfection reagent) | Lipofectamine 3000 | Thermo Fisher | L3000001 | |
| Commercial assay or kit (PCR) | KAPA2G Robust 2 x Hotstart mix | Roche | KK5702 | |

## Plasmid cloning

All crRNA and petracrRNA constructs were cloned using ligation after restriction (T4 DNA Ligase, New England Biolabs), following the protocol outlined in *Anzalone et al., 2019*. Single-stranded DNAs (IDT) were annealed to have 4 bp overhangs in both ends of double-stranded DNAs, which is a substrate for T4 DNA ligase. The plasmid backbone (pU6-pegRNA-GG-acceptor, Addgene_132777) was digested using BsaI-HFv2, and mixed with annealed double-stranded DNA constructs with 4 bp overhangs. At the end of all crRNA and petracrRNA constructs (*Supplementary file 1*), we added the evoPreQ1 sequence and poly-T terminator sequence. A small amount (1–2 µL) of T4 ligation reaction mix was added to NEB Stbl cell (C3040) for transformation and grown at 37°C for the plasmid

DNA preparation (QIAGEN miniprep). The resulting plasmids were sequence-verified using Sanger sequencing (GENEWIZ).

All protein expression constructs (*Supplementary file 1*) were cloned using Gibson Assembly (NEB, where double-stranded DNA fragments are either ordered from IDT as gBlocks or PCR-amplified from existing constructs with at least 25 bp overlap in sequence). A small amount (1–2 µL) of Gibson Assembly reaction mix was added to NEB Stbl cell (C3040) for transformation and grown at 30°C or 37°C for the plasmid DNA preparation (QIAGEN miniprep). The resulting plasmids were sequence-verified using Sanger sequencing (GENEWIZ).

## Tissue culture, transfection, lentiviral transduction, and transgene integration

The HEK293T cell line was purchased from ATCC, and maintained by following the recommended protocol from the vendor. HEK293T cells were cultured in Dulbecco's modified Eagle's medium (DMEM) with high glucose (GIBCO), supplemented with 10% fetal bovine serum (Rocky Mountain Biologicals) and 1% penicillin-streptomycin (GIBCO). Cells were grown with 5% $CO_2$ at 37°C. The HEK293T cell lines were authenticated and tested for mycoplasma using ATCC Cell Line Authentication and Mycoplasma PCR tests.

For transient transfection, HEK293T cells were cultured to 70–90% confluency in a 24-well plate. For prime editing with crRNA/petracrRNA and without protein components, 375 ng of PE4max enzyme plasmid (Addgene_174828), 62.5 ng of crRNA plasmid, and 62.5 ng of petracrRNA plasmid were mixed, and prepared with a transfection reagent (Lipofectamine 3000) following the recommended protocol from the vendor. For transfection with protein components, 250 ng of PE4max enzyme plasmid, 125 ng of protein (MCP-LambdaN, GFP filler plasmid, or 62.5 ng each of two MCP/LambdaN-tagged protein domains), 62.5 ng of crRNA plasmid, and 62.5 ng of petracrRNA plasmid were used in transfection. The transfection mix composition for the experiment with the ADAR-based RNA sensor is described in *Figure 4c*. Cells were cultured for 3–4 days after the initial transfection, and their genomic DNA was harvested following cell lysis and protease protocol from *Anzalone et al., 2019*.

## Genomic DNA collection and sequencing library preparation

The targeted region from collected genomic DNA was amplified using two-step PCR and sequenced using the Illumina sequencing platform (NextSeq). The first PCR (KAPA Robust polymerase) included 1.5 µL of cell lysate, and 0.04–0.4 µM of forward and reverse primers (*Supplementary file 1*) in a final reaction volume of 25 µL. We programmed the first PCR to be: (1) 3 min at 95°C, (2) 15 s at 95°C, (3) 10 s at 65°C, (4) 90 s at 72°C, (5) 25–28 cycles of repeating step 2 through 4, and (6) 1 min at 72°C. Primers included sequencing adapters to their 3'-ends, appending them to both termini of PCR products that amplified genomic DNA. After the first PCR step, products were added to the second PCR that appended dual sample indexes and flow cell adapters. The second PCR program was identical to the first PCR program except we ran it for only 5–10 cycles. Products were purified using AMPure and assessed on the TapeStation (Agilent) before being denatured for the sequencing run.

## Genomic DNA amplicon sequencing data processing and analysis

Sequencing reads from Illumina NextSeq platforms are first demultiplexed using BCL2fastq software (Illumina). Sequencing libraries were single-end sequenced to cover the DNA Tape from one direction. Editing efficiencies were calculated using pattern-matching software such as Regular Expression (package *REGEX*) in Python, counting correct amplicon reads with or without intended edits. For prime editing, CTT insertions to the HEK3 locus (*Anzalone et al., 2019*) were counted, and editing efficiencies were calculated by: reads with CTT insertions/all reads matching the HEK3 locus. In quantifying base editing efficiencies, we first used CRISPResso to identify the top 10 base editing outcomes of the HEK3 locus, and used these editing patterns to calculate editing efficiency as follows: all reads matching the top 10 base editing outcomes/all reads matching HEK3 locus.

## Acknowledgements

We thank members of the Shendure Lab, as well as members of the Allen Discovery Center for Cell Lineage Tracing, for helpful discussions. This work was supported by a grant from the Paul G Allen

Frontiers Group (Allen Discovery Center for Cell Lineage Tracing to JS), the National Human Genome Research Institute (UM1HG011586 and R01HG010632 to JS, and K99HG012973 and R00HG012973 to JC), and the NIH/NCI Cancer Center Support Grant (P30CA008748 supporting JC in part). HL is supported by the NSF Graduate Research Fellowship Program (DGE-2140004). JC is supported by Damon Runyon Cancer Research Fellowship (DRG-2403-20) and Dale Frey Award (DFS-64-24). JS is an Investigator at the Howard Hughes Medical Institute.

## Additional information

### Competing interests

Junhong Choi, Wei Chen: The University of Washington has filed a patent application based on this work, in which J.C., W.C., and J.S. are listed as inventors (WO2024107927A1). The other authors declare that no competing interests exist.

### Funding

| Funder | Grant reference number | Author |
|---|---|---|
| Allen Discovery Center | Cell Lineage Tracing | Jay Shendure |
| National Human Genome Research Institute | UM1HG011586 | Jay Shendure |
| National Human Genome Research Institute | R01HG010632 | Jay Shendure |
| National Human Genome Research Institute | K99HG012973 | Junhong Choi |
| National Human Genome Research Institute | R00HG012973 | Junhong Choi |
| National Science Foundation Graduate Research Fellowship Program | DGE-2140004 | Hanna Liao |
| Damon Runyon Cancer Research Foundation | DRG-2403-20 | Junhong Choi |
| Damon Runyon Cancer Research Foundation | DFS-64-24 | Junhong Choi |
| National Institutes of Health | P30CA008748 | Junhong Choi |

The funders had no role in study design, data collection and interpretation, or the decision to submit the work for publication.

### Author contributions

Junhong Choi, Conceptualization, Data curation, Formal analysis, Funding acquisition, Investigation, Visualization, Methodology, Writing – original draft, Writing – review and editing; Wei Chen, Conceptualization, Methodology, Writing – review and editing; Hanna Liao, Xiaoyi Li, Data curation, Methodology, Writing – review and editing; Jay Shendure, Conceptualization, Supervision, Funding acquisition, Investigation, Writing – original draft, Writing – review and editing

### Author ORCIDs

Junhong Choi ⓘ https://orcid.org/0000-0001-9291-5977
Jay Shendure ⓘ https://orcid.org/0000-0002-1516-1865

Reviewer #1 (Public review): https://doi.org/10.7554/eLife.98110.3.sa1
Reviewer #2 (Public review): https://doi.org/10.7554/eLife.98110.3.sa2
Author response https://doi.org/10.7554/eLife.98110.3.sa3

## Additional files

### Supplementary files

Supplementary file 1. Nucleic acid sequences used in this study.

MDAR checklist

Source data 1. Combined table including source data (editing efficiencies) related to all figures and figure supplements in this article.

### Data availability

Raw sequencing data have been uploaded to Sequencing Read Archive (SRA) with the associated BioProject ID PRJNA1004865. The following plasmids have been deposited to Addgene: pU6-crRNA-MS2, pU6-BoxB-petracrRNA, pCMV-LambdaN-MCP, pCMV-LambdaN-NbALFA, and pCMV-ALFA-MCP (Addgene ID 207624 - 207628). The rest of the plasmids used in this study are available upon request.

The following dataset was generated:

| Author(s) | Year | Dataset title | Dataset URL | Database and Identifier |
|---|---|---|---|---|
| Choi J | 2023 | A dual-RNA-guided molecular proximity sensor that records to genomic DNA | https://www.ncbi.nlm.nih.gov/bioproject/PRJNA1004865 | NCBI BioProject, PRJNA1004865 |

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
