## [Editor Report · eLife Assessment]

This **important** manuscript describes a creative approach using dual-component gRNAs to create a new class of molecular proximity sensors for genome editing. The authors demonstrate that this tool can be coupled with several different gene editing effectors, showing **convincingly** that the tool functions as intended. This study not only introduces a first-of-its kind approach, but through careful measurements also enables future further development of the technology.

---

## [Referee Report · Reviewer #1 (Public review)]

Summary:

The manuscript by Choi and co-authors presents "P3 editing", which leverages dual-component guide RNAs (gRNA) to induce protein-protein proximity. They explore three strategies for leveraging prime-editing gRNA (pegRNA) as a dimerization module to create a molecular proximity sensor that drives genome editing, splitting a pegRNA into two parts (sgRNA and petRNA), inserting self-splicing ribozymes within pegRNA, and dividing pegRNA at the crRNA junction. Among these, splitting at the crRNA junction proved the most promising, achieving significant editing efficiency. They further demonstrated the ability to control genome editing via protein-protein interactions and small molecule inducers by designing RNA-based systems that form active gRNA complexes. This approach was also adaptable to other genome editing methods like base editing and ADAR-based RNA editing.

Strengths:

The study demonstrates significant advancements in leveraging guide RNA (gRNA) as a dimerization module for genome editing, showcasing its high specificity and versatility. By investigating three distinct strategies-splitting pegRNA into sgRNA and petRNA, inserting self-splicing ribozymes within the pegRNA, and dividing the pegRNA at the repeat junction-the researchers present a comprehensive approach to achieving molecular proximity and reconstituting function. Among these methods, splitting the pegRNA at the repeat junction emerged as the most promising, achieving editing efficiencies up to 76% of the control, highlighting its potential for further development in CRISPR-Cas9 systems. Additionally, the study extends genome editing control by linking protein-protein interactions to RNA-mediated editing, using specific protein-RNA interaction pairs to regulate editing through engineered protein proximity. This innovative approach expands the toolkit for precision genome editing, demonstrating the feasibility of controlling genome editing with enhanced specificity and efficiency.

Weaknesses:

The initial experiments with splitting the pegRNA into sgRNA and petRNA showed low editing efficiency, less than 2%. Similarly, inserting self-splicing ribozymes within pegRNA was inefficient, achieving under 2% editing efficiency in all constructs tested, possibly hindered by the prime editing enzyme. The editing efficiency of the crRNA and petracrRNA split at the repeat junction varied, with the most promising configurations only reaching 76% of the control efficiency. The RNA-RNA duplex formation's inefficiency might be due to the lack of additional protein binding, leading to potential degradation outside the Cas9-gRNA complex. Extending the approach to control genome editing via protein-protein interactions introduced complexity, with a significant trade-off between efficiency and specificity, necessitating further optimization. The strategy combining RADARS and P3 editing to control genome editing with specific RNA expression events exhibited high background levels of non-specific editing, indicating the need for improved specificity and reduced leaky expression. Moreover, P3 editing efficiencies are exclusively quantified after transfecting DNA into HEK cells, a strategy that has resulted in past reproducibility concerns for other technologies. Overall, the various methods and combinations require further optimization to enhance efficiency and specificity, especially when integrating multiple synthetic modules.

Comments on revisions:

I think the authors have successfully addressed the initial concerns. Their adaption of the main text and discussion makes the limitations of P3 editing much clearer.

---

## [Referee Report · Reviewer #2 (Public review)]

Choi et al. describes a new approach for enabling input-specific CRISPR-based genome editing in cultured cells. While CRISPR-Cas9 is a broadly applied system across all of biology, one limitation is the difficulty in inducing genome editing based on cellular events. A prior study, from the same group, developed ENGRAM - which relies on activity-dependent transcription of a prime editing guide RNA, which records a specific cellular event as a given edit in a target DNA "tape". However, this approach is limited to detection of induced transcription, and does not enable the detection of broader molecular events including protein-protein interactions or exposure to small molecules. As an alternative, this study envisioned engineering the reconstitution of a split prime editing guide RNA (pegRNA) in a protein-protein interaction (PPI)-dependent manner. This would enable location- and content-specific genome editing in a controlled setting.

Strengths:

The strengths of this paper include an interesting concept for engineering guide RNAs to enable activity-dependent genome editing in living cells in the future, based on discreet protein-protein interactions (either constitutively, spatially, or chemically induced). Important groundwork is laid down to engineer and improve these guide RNAs in the future (especially the work describing altering the linkers in Supplementary Figure 3 - which provides a path forward).

Weaknesses:

In its current state, the editing efficiency appears too low to be applied in physiological settings. Much of the latter work in the paper relies on a LambdaN-MCP direction fusion protein, rather than two interacting protein pairs. Further characterizations in the future, especially varying the transfection amounts/durations/etc of the various components of the system, would be beneficial to improve the system. It will also be important to demonstrate editing at additional sites; to characterize how long the PPI must be active to enable efficient prime editing; and how reversible the reconstitution of the split pegRNA is.

In the revised version, the authors clearly describe the present limitations of the system in the discussion section, and also highlight specific actions and potential approaches for improving the efficiency of the system for application in biological systems. They also add further insight into why it is advantageous to design engineered guideRNAs, as opposed to engineered Cas9 enzymes, to improve the modularity of the system in the future.

---

## [Author Response]

The following is the authors’ response to the original reviews.

**Public Reviews:**

**Reviewer #1 (Public Review):**
Summary:The manuscript by Choi and co-authors presents "P3 editing", which leverages dual-component guide RNAs (gRNA) to induce protein-protein proximity. They explore three strategies for leveraging prime-editing gRNA (pegRNA) as a dimerization module to create a molecular proximity sensor that drives genome editing, splitting a pegRNA into two parts (sgRNA and petRNA), inserting self-splicing ribozymes within pegRNA, and dividing pegRNA at the crRNA junction. Among these, splitting at the crRNA junction proved the most promising, achieving significant editing efficiency. They further demonstrated the ability to control genome editing via protein-protein interactions and small molecule inducers by designing RNA-based systems that form active gRNA complexes. This approach was also adaptable to other genome editing methods like base editing and ADAR-based RNA editing.Strengths:The study demonstrates significant advancements in leveraging guide RNA (gRNA) as a dimerization module for genome editing, showcasing its high specificity and versatility. By investigating three distinct strategies-splitting pegRNA into sgRNA and petRNA, inserting self-splicing ribozymes within the pegRNA, and dividing the pegRNA at the repeat junction-the researchers present a comprehensive approach to achieving molecular proximity and reconstituting function. Among these methods, splitting the pegRNA at the repeat junction emerged as the most promising, achieving editing efficiencies up to 76% of the control, highlighting its potential for further development in CRISPR-Cas9 systems. Additionally, the study extends genome editing control by linking protein-protein interactions to RNA-mediated editing, using specific protein-RNA interaction pairs to regulate editing through engineered protein proximity. This innovative approach expands the toolkit for precision genome editing, demonstrating the feasibility of controlling genome editing with enhanced specificity and efficiency.Weaknesses:The initial experiments with splitting the pegRNA into sgRNA and petRNA showed low editing efficiency, less than 2%. Similarly, inserting self-splicing ribozymes within pegRNA was inefficient, achieving under 2% editing efficiency in all constructs tested, possibly hindered by the prime editing enzyme. The editing efficiency of the crRNA and petracrRNA split at the repeat junction varied, with the most promising configurations only reaching 76% of the control efficiency. The RNA-RNA duplex formation's inefficiency might be due to the lack of additional protein binding, leading to potential degradation outside the Cas9-gRNA complex. Extending the approach to control genome editing via protein-protein interactions introduced complexity, with a significant trade-off between efficiency and specificity, necessitating further optimization. The strategy combining RADARS and P3 editing to control genome editing with specific RNA expression events exhibited high background levels of non-specific editing, indicating the need for improved specificity and reduced leaky expression. Moreover, P3 editing efficiencies are exclusively quantified after transfecting DNA into HEK cells, a strategy that has resulted in past reproducibility concerns for other technologies. Overall, the various methods and combinations require further optimization to enhance efficiency and specificity, especially when integrating multiple synthetic modules.

Thank you for this accurate summary and assessment of the strengths and weaknesses of the P3 editing as it stands. Looking ahead, we agree that further optimizations will be important, as will characterizing the performance of P3 editing in additional cellular contexts. The revised Discussion (see below) now makes these points more clearly.

**Reviewer #2 (Public Review):**
Choi et al. describe a new approach for enabling input-specific CRISPR-based genome editing in cultured cells. While CRISPR-Cas9 is a broadly applied system across all of biology, one limitation is the difficulty in inducing genome editing based on cellular events. A prior study, from the same group, developed ENGRAM - which relies on activity-dependent transcription of a prime editing guide RNA, which records a specific cellular event as a given edit in a target DNA "tape". However, this approach is limited to the detection of induced transcription and does not enable the detection of broader molecular events including protein-protein interactions or exposure to small molecules. As an alternative, this study envisioned engineering the reconstitution of a split prime editing guide RNA (pegRNA) in a protein-protein interaction (PPI)-dependent manner. This would enable location- and content-specific genome editing in a controlled setting.The authors explored three different design possibilities for engineering a PPI-dependent split pegRNA. First, they tried splitting pegRNA into a functional sgRNA and corresponding prime editing transRNA, incorporating reverse-complementary dimerization sequences on each guide half. This approach, however, resulted in low editing efficiency across 7 different designs with various complementary annealing template lengths (<2% efficiency). They also tried inserting a self-splicing ribozyme within the pegRNA, which produces a functional pegRNA post-transcriptionally. The incorporation of a split-ribozyme, dependent on a PPI, could have been used to reconstitute the split pegRNA in an event-controlled manner. However again, only modest levels of editing were observed with the self-splicing ribozyme design (<2%). Finally, they tried splitting the pegRNA at the repeat:anti-repeat junction that was used to join the original dual-guide system comprised of a crRNA and tracrRNA, into a single-guide RNA. They incorporated the prime editing features into the tracrRNA half, to create petracrRNA. Dimerization was initially induced by different complementary RNA annealing sequences. Using this design, they were able to induce an editing efficiency of ~28% (compared to 37% efficiency using a positive control epegRNA guide).Having identified a suitable split pegRNA system, they next sought to induce the reconstitution of the two halves in a PPI-dependent manner. They replaced the complementary RNA annealing sequences with two different RNA aptamers (MS2 and BoxB). MS2 detects the MCP protein, while BoxB detects the LambdaN protein. Close proximity between MCP and LambdaN would thus bring together the two split pegRNA halves, creating a functional pegRNA that would enable prime editing at a specific target site. They demonstrated that they could induce MCP-BoxB proximity by fusing them to different dimerizing protein partners: (1) constitutive epitope-nanobody/antibody pairs such as scFv/GCN4 or NbALFA/ALFA-Tag; (2) split-GFP; or (3) chemically-induced protein pairs such as FKBP/FRB or ABI/PYL. For all of these approaches, they could achieve between ~20-60% normalized editing efficiency (relative to positive control editing levels with epegRNA). Additional mutation of the linkers between the RNA and aptamers could increase editing efficiency but also increase non-specific background editing even in the absence of an induced PPI.Additional applications of this overall strategy included incorporating the design with different DNA base editors, with the most promising examples shown with the base editors CBE4max and ABE8. It should be noted that these specific examples used a non-physiological LambdaN-MCP direct fusion protein as the "bait" that induced reconstitution of the two halves of the guideRNA, rather than relying on a true induced PPI. They also demonstrated that the recently reported RADARS strategy could be incorporated into their system. In this example, they used an ADAR-guide-RNA to drive the expression of a LambdaN-PCP fusion protein in the presence of a specific target RNA molecule, IL6. This induced LambdaN-PCP protein could then reconstitute the split peg-RNAs to drive prime editing. To enable this last application, they replaced the MS2 aptamer in their pegRNA with the PP7 aptamer that binds the PCP protein (this was to avoid crosstalk with RADARS, which also uses MS2/MCP interaction). Using this strategy, they observed a normalized editing efficiency of around 12% (but observed non-specific editing of around 8% in the absence of the target RNA).Strengths:The strengths of this paper include an interesting concept for engineering guide RNAs to enable activity-dependent genome editing in living cells in the future, based on discreet protein-protein interactions (either constitutively, spatially, or chemically induced). Important groundwork is laid down to engineer and improve these guide RNAs in the future (especially the work describing altering the linkers in Supplementary Figure 3 - which provides a path forward).Weaknesses:In its current state, the editing efficiency appears too low to be applied in physiological settings. Much of the latter work in the paper relies on a LambdaN-MCP direction fusion protein, rather than two interacting protein pairs. Further characterizations in the future, especially varying the transfection amounts/durations/etc of the various components of the system, would be beneficial to improve the system. It will also be important to demonstrate editing at additional sites; to characterize how long the PPI must be active to enable efficient prime editing; and how reversible the reconstitution of the split pegRNA is.

Thank you for this assessment of the strengths and weaknesses of the P3 editing as it stands. Looking ahead, we agree that further optimizations will be important, including along the lines suggested by the reviewer, as will further characterization of the system with respect to dependencies, reversibility, etc. The revised Discussion (see below) now makes these points more clearly.

**Recommendations for the authors:**

**Reviewing Editor comments:**
It would be helpful to better describe the nature of improvements (on-targeting and/or off-targeting) that would be needed to effectively use this approach in vitro and in vivo applications.

We agree, and have accordingly revised the last paragraph of our discussion to better describe what improvements are needed for in vitro and in vivo applications:

“In our view, there are four outstanding challenges for P3 editing to be broadly useful: evaluating additional cellular contexts, the method’s efficiency and specificity, understanding the limit of detectable protein-protein interactions, and the development of sensors compatible with multiplex P3 editing within the same cell. First, we have thus far only conducted P3 editing in HEK293T cells, and obviously needs to be tested in additional cell types. Second, both the efficiency and specificity of the P3 editing need to be improved before it can be used as a selective editing tool in model systems. We have explored how modifying the crRNA and petracrRNA pair sequences can tune the efficiency-vs-specificity tradeoff, but alternative avenues to improvement (e.g., better docking of RNA-aptamers such as MS2, BoxB, or PP7 by testing more linker sequences that place crRNA and petracrRNA for duplex formation) may be more fruitful in terms of achieving high efficiency and specificity at once (e.g., >50% editing in the setting of a specific protein-protein interaction, and <1% editing without it). Second, it is not clear whether weak and transient interactions among proteins can be used to trigger P3 editing. Assuming the genome editing complex formation is reversible, improving P3 editing efficiency may be able to capture different strengths of protein-protein interactions, although some interactions may be too transient to promote functional guide RNA formation. Finally, the current P3 editing design uses a pair of RNA aptamers and their corresponding protein binders, limiting the multiplex detection of protein-protein pairs. More orthogonal protein-RNA pairs need to be identified (e.g., using a massively parallel platform (Buenrostro et al., 2014) and/or computational prediction (Baek et al., 2023)) to allow for large numbers of P3 sensors for different protein-protein interactions to be deployed within the same cell. Overcoming these four challenges is necessary for P3 editing to be broadly useful for gating genome editing on physiological levels of specific protein-protein interactions in a multiplex fashion.”

**Reviewer #2 (Recommendations For The Authors):**
It does not appear that all plasmids necessary to reproduce the results of this paper have been deposited to addgene, but only a small subset. The authors might include that these plasmids are available upon request, if not uploaded to a public repository.

We have added a statement that additional plasmids are available upon request. Our Data Availability Statement reads (with the added sentence underlined):

“Raw sequencing data have been uploaded to Sequencing Read Archive (SRA) with the associated BioProject ID PRJNA1004865. The following plasmids have been deposited to Addgene: pU6-crRNA-MS2, pU6-BoxB-petracrRNA, pCMV-LambdaN-MCP, pCMV-LambdaN-NbALFA, and pCMV-ALFA-MCP (Addgene ID 207624 - 207628). The rest of the plasmids used in this study are available upon request.”

It could be useful to include somewhere why, specifically, editing the guide RNAs as opposed to the Cas9 itself is advantageous. Light-inducible split Cas9s have been engineered, and I imagine other PPI-inducible split Cas9s have also been engineered. A specific mention of the advantages of using engineered split pegRNAs could put the significance of this work in a better context.

Thanks for raising this, and we agree. We have revised the first paragraph of the Results section to highlight why we think splitting the guide RNAs as opposed to Cas9 might be advantageous:

“In the split architecture, the “dimerization module” is a key sensor component. Although strategies that split the protein component of the genome editing complex have been described (e.g., split-Cas9 (Yu et al., 2020)), we reasoned that having the guide RNA serve as the dimerization module rather than the protein, *i.e.* by splitting it into two parts, and making the restoration of its function dependent on a molecular proximity event, would afford even more control. For example, if multiple split gRNAs were present within the same cell, they could be independently controlled, whereas a split Cas9 would only allow a single control point. In our initial experiments, we focused on splitting the pegRNA used in prime editing.”